# Association between Drug Use and Perception of Mental Health in Women Diagnosed with Fibromyalgia: An Observational Study

**DOI:** 10.3390/biomedicines12102284

**Published:** 2024-10-09

**Authors:** Andrea Lizama-Lefno, Krystel Mojica, Ángel Roco-Videla, Juan Ignacio Vargas Ruiz-Tagle, Nelia González-Droguett, María Jesús Muñoz-Yánez, Erick Atenas-Núñez, Nelson Maureira-Carsalade, Sergio Flores Carrasco

**Affiliations:** 1Dirección de Desarrollo y Postgrados, Universidad Autónoma de Chile, Galvarino Gallardo 1983, Santiago 7500138, Chile; andrea.lizama@cloud.uautonoma.cl; 2Fundación Núcleo de Investigación DOLMEN, El Director 6000, Of. 207, Las Condes, Santiago 7580023, Chile; krystel.mojica@fundaciondolmen.org (K.M.); nelia.gonzalez@fundaciondolmen.org (N.G.-D.); 3Vicerrectoría de Investigación e Innovación, Universidad Arturo Prat, Iquique 1110939, Chile; anroco@unap.cl; 4Clínica Andes Salud, Bellavista 123, Puerto Montt 5480000, Chile; drjuanignaciovargas@gmail.com; 5Universidad Gabriela Mistral, Av. Andrés Bello 1337, Santiago 7500533, Chile; mariajesus.munoz@ugm.cl (M.J.M.-Y.); erick.atenas.n@ugm.cl (E.A.-N.); 6Facultad de Ingeniería, Universidad Católica de la Santísima Concepción, Concepción 4090541, Chile; nmaureira@ucsc.cl; 7Instituto de Geografía, Universidad Católica de Valparaíso, Valparaíso 2340000, Chile

**Keywords:** fibromyalgia, mental health, sleep quality, medication therapy management, drug prescriptions, benzodiazepines, zolpidem, serotonin–norepinephrine reuptake inhibitors (SNRIs), polypharmacy, women’s health

## Abstract

Background/Objectives: Fibromyalgia (FM) is a chronic syndrome characterized by widespread musculoskeletal pain, fatigue, sleep disturbances, and mental health issues. It affects approximately 1.78% of the general population; an estimated 4:1 ratio between women and men is observed. It significantly impacts quality of life and carries both clinical and social stigma. This study aims to evaluate the relationship between drug use and mental health in female patients with fibromyalgia. Methods: This study is prospective, observational, and cross-sectional. A questionnaire was administered to 544 subjects, achieving a representative sample size from a population of 800,000 subjects by using an algorithm for proportion estimation with a known sampling frame. The selection was non-random, making the sampling non-probabilistic. Logistic regression models were applied to assess the effect of drug use on perception of mental health; presence of symptoms such as comprehension and memory problems, insomnia, depression, and anxiety; and severity of cognitive symptoms and non-restorative sleep. To quantify the impact, odds ratios and confidence intervals have been observed. Results: The findings indicate the non-recommended use of medications and reveal the ineffectiveness and adverse effects of drug interactions on mental health. The use of benzodiazepines and sedative-hypnotics is significantly associated with a negative perception of mental health. Benzodiazepines do not improve symptoms or significantly reduce their severity. SSRI antidepressants do not enhance mental health perception; however, when used exclusively, they are effective in reducing the severity, but not the prevalence, of cognitive symptoms. Conclusions: The results highlight the complexity of pharmacological management in FM and raise concerns about the inappropriate use of ineffective or counterproductive drug interactions affecting patients’ mental health. They underscore the need for multidisciplinary and personalized strategies that include close and careful monitoring, as well as the simultaneous use of non-pharmacological treatments that have demonstrated evidence in improving quality of life without negatively affecting mental health, such as patient education, psychological therapy, physiotherapy, and mindfulness.

## 1. Introduction

Fibromyalgia (FM) is a chronic syndrome characterized by widespread musculoskeletal pain, fatigue, sleep disturbances, and cognitive symptoms, often associated with mental disorders such as depression and anxiety. It affects approximately 1.78% of the general population, predominantly affecting women, who are up to four times more likely to be diagnosed compared to men, with prevalence rates in women ranging from 2% to 6%, and it is most commonly diagnosed between the ages of 20 and 55 [1]. The management of FM presents significant challenges due to the incomplete understanding of its etiology and the limited availability of scientifically proven therapies. Central sensitization is considered a key pathophysiological mechanism in the development of fibromyalgia. However, factors such as distress, dysautonomia, and neuroinflammation are also thought to promote central sensitization, contributing to chronic pain and symptom variability in fibromyalgia [2].

Emotional distress can exacerbate FM symptoms, making it critical to include psychological assessments in treatment. Dysautonomia can lead to fatigue and sleep problems; a balanced diet and specific exercises that promote autonomic function are recommended to help mitigate some symptoms. Neuroinflammation, primarily driven by glial hyperactivity, is likely a key factor in the initiation and maintenance of central sensitization. It is also linked to other conditions commonly associated with chronic pain, such as depression, sleep disorders, and anxiety [3]. As a result, efforts are underway to develop therapeutic strategies that modulate specific inflammatory pathways and glial activation mediators to address both chronic pain and its related comorbidities [4]. Ultimately, this highlights the need for multimodal approaches that address both neuronal and psychological factors [5].

Many therapeutic approaches focus on various mechanisms associated with the process of central sensitization, such as temporal summation, conditioned modulation, and dysfunction of descending inhibition [6]. Serotonin and norepinephrine reuptake inhibitors (SNRIs) and GABAergic drugs, both FDA-approved and recommended by the European League Against Rheumatism (EULAR), modify key mechanisms associated with this central sensitization [5,7]. SNRIs act on the descending pathways, while GABAergic drugs reduce the excitability of the central nervous system, helping to regulate pain related to central sensitization [8].

SNRIs increase the availability of serotonin and norepinephrine in the descending inhibitory pathways, which decreases the central amplification of pain signals and improves mood, addressing comorbidities such as depression and anxiety in patients with FM [5]. Pregabalin, by modulating calcium channels in the central nervous system, reduces neuronal hyperexcitability involved in pain transmission [5,7]. Off-label treatments include amitriptyline, mirtazapine, cyclobenzaprine, gabapentin, and naltrexone [8]. The neuroanatomical circuits involved in pain, emotion, and motivation are closely interconnected, suggesting that the modulation of neurotransmitters like serotonin and norepinephrine can simultaneously regulate pain processing pathways and alleviate mood disorders frequently observed in FM [9]. On the other hand, various drugs that act peripherally, such as non-steroidal anti-inflammatory drugs (NSAIDs) and COX-2 inhibitors, are not commonly considered in conditions of central sensitization like fibromyalgia [6].

The variability in adherence to prescribed treatment guidelines for FM is evident, with high rates of treatment discontinuation and self-medication observed [10,11]. Many patients resort to non-recommended medications such as opioids and NSAIDs despite the lack of solid evidence supporting their use. A Canadian study found that 33% of FM patients used opioids, 54% used NSAIDs, and 23.8% used tramadol. Among the recommended medications, 36.5% used anticonvulsants, 55.6% used SNRIs, and 22.2% used tricyclic antidepressants [10]. This variability is often due to side effects, complex regimens, and the perceived lack of efficacy of current therapies. These findings highlight the need for personalized treatment approaches and close monitoring to improve adherence and patient outcomes in FM management.

A previous report indicates that the three most commonly used drug families are GABAergic and anticonvulsants (pregabalin, gabapentin, or carbamazepine), SNRI antidepressants, and analgesics (paracetamol and/or tramadol). Approximately 77.3% of patients are exposed to polypharmacy, with an average of 2.8 drugs used per patient [11]. Although medications can provide symptomatic relief for patients with FM, their long-term effectiveness is limited, and they should be administered with caution due to potential adverse effects.

The complexity of pharmacological treatment in FM has led to the exploration of new therapeutic options, such as LSD, psilocybin, cannabinoids, and vitamin D supplementation, although more evidence is needed to support their use [12,13]. Recent reviews suggest that combined therapeutic approaches, incorporating both pharmacological and nonpharmacological interventions, are the most effective strategies for managing FM. Non-pharmacological interventions, particularly physical exercise, health education, and cognitive-behavioral psychological techniques, are the ones with the greatest scientific support [14,15,16], as well as dietary interventions and relaxation techniques [17,18,19,20,21,22,23].

Understanding the impact of medication on mental health can help address issues of treatment efficacy and adherence in FM. Identifying drug interactions that worsen mental health could contribute to more precise prescriptions, lead to more personalized treatments, minimize adverse effects, and consequently improve adherence and therapeutic outcomes, thereby enhancing mental well-being. The present study aims to evaluate the relationship between medication use and mental health in female patients with FM, specifically analyzing the impact of medication use and its interactions on self-perceived mental health, symptom presence, and symptom severity.

## 2. Materials and Methods

### 2.1. Type of Study

A quantitative, observational, prospective, cross-sectional study was conducted.

### 2.2. Instrument and Variables

A self-administered online questionnaire with 60 questions was used, encompassing 93 variables distributed across eight sections that contribute to the dimensions of the biopsychosocial approach, including general health status, physical and mental health status, specific symptoms, comorbidities and treatments, mental health and psychological well-being, sociodemographic characteristics, work-related aspects, and social and family support. The questions explored multiple physical symptoms associated with FM, primarily pain and fatigue; psychological factors such as sleep quality, cognitive capacity, anxiety, and depression; and social elements, including family dynamics and social recognition. This comprehensive approach was designed to capture the multifaceted impact of fibromyalgia on patients’ lives, ensuring relevance and clarity for the target population.

This article analyzed three dependent variables: perception of mental health (measured on a scale from 1 to 5), presence of symptoms (presence/absence of comprehension and memory problems, insomnia, depression, and anxiety), and severity of cognitive symptoms and non-restorative sleep (measured on a scale from 1 to 5). One independent variable was studied: drug use (use/non-use of GABAergic and anticonvulsants, SNRI antidepressants, analgesics, NSAIDs, benzodiazepines, muscle relaxants, Zolpidem/Zopiclone, tricyclic antidepressants, ergotamine Migranol, methotrexate, quetiapine, other, no drug use).

The item assessing the presence of symptoms included the 40 symptoms evaluated by the Symptom Severity Score (SS Score) and their degree of severity measured using the same instrument [24]. One item inquired about the medications currently being used, and responses were subsequently grouped into drug families. The questionnaire was validated using the Delphi method by a multidisciplinary panel of eight experts in the field, drawn from medical and social sciences. Two rounds of the Delphi method were applied: In the first round, experts provided feedback on the relevance, clarity, appropriateness, and comprehensiveness of the questions, adding questions to better capture each dimension, ensuring that the instrument adequately covered all areas intended to be measured. The second round focused on eliminating ambiguity, reducing complexity, and improving comprehensibility. Key changes included rephrasing items, refining the wording, and ensuring that all items were easily understood by the target population. The overall structure of the instrument was evaluated, including the order of the items and the coherence between sections.

Subsequently, a pretest was conducted with 26 subjects, who completed the questionnaire and then filled out a brief form in which they evaluated the clarity, length, and comprehensibility of the instrument. They were asked to indicate the presence of ambiguous questions or difficulties in answering the items, comment on the comfort with the language and format, and provide suggestions for improvement. Based on the feedback obtained, some adjustments were made to the instrument before its implementation.

### 2.3. Sampling

An estimated 800,000 people in Chile have FM. The sample size was calculated using an algorithm for proportion estimation with a known sampling frame, a sampling error of 1%, a confidence level of 99%, a prevalence of 3%, and a margin of error of 2%. An open call for participation in the study was made through the social networks of two organizations associated with FM, providing a virtual link to the questionnaire. Upon clicking the link, participants were required to read and sign the online informed consent form before the questionnaire was displayed. A total of 663 questionnaires were collected, of which 544 cases involving women who met the criteria of being diagnosed with FM, being over 18 years of age, and having signed the informed consent form were validated, resulting in a non-probabilistic sample representative in size.

A non-probabilistic sample was chosen due to the challenges in reaching the fibromyalgia community, including variable diagnosis and dispersed locations. This approach facilitated recruitment through support groups and clinics but may introduce selection bias, possibly overrepresenting those actively seeking care. Despite this, the method aimed to capture diverse patient experiences.

Participants were recruited from two fibromyalgia organizations, where membership requires a confirmed diagnosis by a healthcare professional, such as a rheumatologist. This indirect verification of diagnosis strengthens the reliability of the sample, ensuring that responses are from individuals with a clinical diagnosis of fibromyalgia.

### 2.4. Data Analysis

Data were exported from Google Forms in “.xlsx” format, anonymized, and imported into PASW 18.0 in “.sav” format. Descriptive, bivariate, and multivariate analyses were performed. Categorical data were described using frequency and proportion analysis. Bivariate analysis was conducted using the χ^2^ test and cross-tabulation, followed by binary and ordinal logistic regression models to assess the effect of drug use on the dependent variables. A confidence level of 95% (*p* < 0.05) was used.

### 2.5. Ethical Considerations

This study was conducted in accordance with the ethical standards set by the Declaration of Helsinki and received approval from the Institutional Review Board of the University of Santiago. Participants provided informed consent, ensuring their voluntary participation and the confidentiality of their data. Given the sensitive nature of the study involving human subjects, all procedures were designed to minimize potential risks and ensure the ethical handling of personal information.

To ensure data security, all collected responses were stored in encrypted databases with restricted access, available only to the principal investigators. Personal identifiers were removed to maintain participant anonymity, and all data handling procedures complied with institutional and national regulations on data protection.

## 3. Results

### 3.1. Description of Perceived Mental Health

The mean age of the patients studied was 44 years (SD = 10.86), with 64.1% being between 35 and 55 years old. Regarding their perception of mental health over the last year, 72.6% rated it as poor or fair, while 27.4% considered their mental health to be good, very good, or excellent. Among the symptoms associated with mental health, the most prevalent were comprehension or memory problems, followed by insomnia and depression, with anxiety being the least prevalent. The perceived severity of sleep quality (non-refreshing sleep) and cognitive symptoms was also evaluated. For non-refreshing sleep, 85% of participants perceived it as moderate or severe, with nearly 40% rating it as severe. In contrast, the perceived severity of cognitive symptoms was lower, with 89.2% of participants rating them as mild or moderate (Table 1).

### 3.2. Influence on the Perception of Mental Health

Significant relationships between the most common pharmacological patterns and subjective variables—perception of mental health, perceived symptoms, and their severity—were explored. An ordinal logistic regression was conducted to evaluate the influence of medication use on the perception of mental health. Additionally, a generalized linear model was used to quantify the odds ratios for the variables. The assumptions of linearity in the logit for continuous variables were checked by including interaction terms with their natural logarithms, and multicollinearity was assessed using the Variance Inflation Factor (VIF), ensuring acceptable values below 10. The model was statistically significant (X^2^ = 51.338, *p* < 0.05, R^2^ = 0.101). The use of benzodiazepines and Zolpidem/Zopiclone was found to predict mental health perception. Specifically, the use of benzodiazepines reduced the likelihood of having a good perception of mental health (OR = 2.8, 95% CI: 1.82–4.20), and the use of Zolpidem/Zopiclone reduced this likelihood (OR = 2.5, 95% CI: 1.47–4.35) (Table 2). The remaining drugs included in the model did not significantly influence the perception of mental health (*p* > 0.05).

Of the 120 patients using benzodiazepines, 87.5% reported that their mental health was poor or fair. Among the 24 patients using both benzodiazepines and Zolpidem/Zopiclone simultaneously, this percentage was 83.3%. When SSRI antidepressants were also used, the percentage of patients with a poor or fair perception of their mental health increased to 88.2% (Table 3).

### 3.3. Influence on Symptomatic Presence

To evaluate the effect of drug use on the presence of symptoms, a logistic regression analysis was performed. The model was statistically significant (χ^2^ = 29.486, *p* < 0.05, R^2^ = 0.195) (Table 4).

#### 3.3.1. Cognitive Symptoms

The use of GABAergic and anticonvulsants (pregabalin, gabapentin, or carbamazepine) and atypical antipsychotics (quetiapine or risperidone) was associated with the presence of cognitive symptoms (comprehension and memory problems). The use of GABAergic and anticonvulsants increased the likelihood of experiencing cognitive symptoms (OR = 4.016, 95% CI: 1.30–12.45). Among the 290 patients using GABAergic and anticonvulsants, 98.3% presented with cognitive symptoms. Furthermore, eight out of ten patients using atypical antipsychotics experienced these symptoms, and all patients (six subjects) who used both drug types simultaneously reported cognitive problems.

#### 3.3.2. Insomnia

The use of analgesics (paracetamol and/or tramadol), non-steroidal anti-inflammatory drugs (NSAIDs), and benzodiazepines predicted the presence of insomnia (χ^2^ = 33.184, *p* < 0.05, R^2^ = 0.091). Among patients using analgesics, 84.3% reported insomnia. For those using NSAIDs, the prevalence was 86.5%, and for benzodiazepine users, it was 89.2%. Insomnia was observed in 93.9% of the patients using all three drug types simultaneously.

#### 3.3.3. Depression

In the case of depression, the use of analgesics (paracetamol and/or tramadol) and benzodiazepines was a significant predictor (χ^2^ = 25.772, *p* < 0.05, R^2^ = 0.068). The prevalence of depression was 79.1% among analgesic users and 84.2% among benzodiazepine users. The simultaneous use of both drugs was associated with a depression prevalence of 85.3%.

#### 3.3.4. Anxiety

For anxiety, the only drug with a significant influence was the benzodiazepine family. Among patients using benzodiazepines, 65% presented with anxiety symptoms. The logistic regression model was statistically significant (χ^2^ = 22.848, *p* < 0.05, R^2^ = 0.055). None of the other drugs included in the model predicted the presence of the symptoms studied.

### 3.4. Influence on Symptom Severity

An ordinal logistic regression analysis was conducted to assess the influence of drug use on the severity of cognitive symptoms and non-restorative sleep. The model was statistically significant (χ^2^ = 31.728, *p* < 0.05, R^2^ = 0.064). The use of SSRI antidepressants and benzodiazepines was found to predict the severity of cognitive symptoms. The percentage of patients perceiving severe symptoms is highest with the simultaneous use of benzodiazepines and SSRI antidepressants (20.3%). This percentage decreases to 13.6% with the use of benzodiazepines without SSRI antidepressants and further decreases to 9.3% when SSRI antidepressants are used without benzodiazepines. The lowest percentage of perceived severity is observed when SSRI antidepressants are used exclusively (4.3%). In this latter group, 56.5% of patients reported their cognitive symptoms as mild (Table 5). However, the prevalence of cognitive symptoms (comprehension and memory problems) in patients exclusively using SSRI antidepressants remains high at 91.7%.

In the case of non-refreshing sleep, benzodiazepines are the only drugs in the model that have a significant influence. It was statistically significant (χ^2^ = 24.134, *p* < 0.05, R^2^ = 0.049). In patients using benzodiazepines, the perceived severity rate is approximately 50%, both in exclusive use and in interaction with other drugs (Table 6).

## 4. Discussion

The findings indicate off-label use of medications [25,26] and reveal the ineffectiveness and counterproductive effects of drug interactions on the mental health of women with FM. The analysis confirms that the use of benzodiazepines and sedative-hypnotics, such as Zolpidem or Zopiclone, is significantly associated with a negative perception of mental health. Benzodiazepines in particular have been identified as the strongest predictor of poor mental health in FM patients. However, they do not demonstrate effectiveness in improving symptoms or significantly reducing symptom severity.

The strong association between benzodiazepine use and a negative perception of mental health may reflect either a direct effect of these medications on mental health or the possibility that patients with poorer mental health are more likely to be prescribed these drugs, as benzodiazepines are commonly used to treat symptoms of anxiety and other disorders associated with fibromyalgia.

The negative impact of benzodiazepines on mental health in fibromyalgia may be linked to their modulation of GABAergic pathways, which can lead to cognitive impairment and depressive symptoms by suppressing neural activity. Psychologically, their sedative effects can worsen fatigue and decrease motivation, further affecting mental health perception. Dependence and withdrawal symptoms also contribute to a cycle of amplified anxiety. Understanding these mechanisms emphasizes the need for personalized treatment strategies [10].

These findings underscore the importance of examining the contexts in which these medications are prescribed, considering potential gender biases, and reconsidering their use, alongside other hypnotics, in the treatment of fibromyalgia. Additionally, close monitoring of patients who use them is necessary.

In conjunction with benzodiazepines and Zolpidem/Zopiclone, SSRI antidepressants do not improve the perception of mental health; however, when used exclusively, they are effective in reducing the severity, but not the prevalence, of cognitive symptoms. To develop effective pharmacological protocols for fibromyalgia that do not adversely affect mental health, it is critical to study the interactions between benzodiazepines, Zolpidem/Zopiclone, and other drugs. This would help identify treatment regimens associated with the subset of users who perceive their mental health as good or very good, thus contributing to optimizing the prescription of these drugs and reducing harm due to lack of knowledge. However, it is anticipated that the effectiveness of such a regimen may involve a high level of polypharmacy. Furthermore, there is no solid evidence supporting the simultaneous use of multiple drugs [25,26].

Regarding symptomatic presence, the GABAergic and anticonvulsant family (e.g., pregabalin) is one of the three groups approved by the FDA for fibromyalgia treatment and is the most commonly used medication among the studied population [11]. It is primarily prescribed to relieve neuropathic pain and improve symptoms such as sleep disturbances and fatigue. The results show that while these medications have a significant effect on cognitive symptoms, they are ineffective in reducing comprehension and memory problems and have no significant effect on insomnia, depression, or anxiety.

Analgesics, NSAIDs, and benzodiazepines were associated with a significant but ineffective impact on insomnia improvement. These medications are not recommended by the American College of Rheumatology due to their side effects [25,26] and are not FDA-approved for FM treatment [7]. Nevertheless, they are among the most commonly used drugs in the study population [11]. Similar effects were observed with the use of analgesics and benzodiazepines for depression and anxiety. SSRI antidepressants did not show a significant relationship with the presence of mental health symptoms in the model. However, it is important to interpret these results cautiously, as the doses of SSRIs used are not known, and some studies have shown moderate or low significant effects [27,28,29,30].

It is prudent to consider that the significant effects of drug use on symptom presence are framed within multivariate exploratory models. The exact pharmacological interactions that lead to the presence or absence of symptoms are not precisely known. Therefore, the ineffectiveness of certain drugs in reducing mental health symptoms could be influenced by the simultaneous use of other drugs or by the doses administered [31]. Additionally, prescriptions may not necessarily aim to reduce the specific symptoms studied. It would be worthwhile to investigate whether the use of drugs that do not improve mental health significantly alleviates other symptoms, such as pain. For instance, a meta-analysis found that duloxetine was the most effective FDA-approved drug for improving pain and depression, while amitriptyline, a tricyclic antidepressant not approved for FM, was the most effective for improving sleep, fatigue, and patient quality of life [32]. Therefore, the exploration conducted in this analysis does not conclude with the findings presented here but rather raises alerts, questions, and challenges for both research and clinical practice by confirming the ineffectiveness of pharmacological treatments for a complex and multisymptomatic disease like FM.

While the results clearly indicate an adverse relationship between medication use and mental health in fibromyalgia patients, further analysis is necessary to establish conclusive pharmacological protocols, considering multiple symptomatic correlations [16,33] and the distinctly different patient profiles. This highlights the importance of personalizing and optimizing both pharmacological and psychological treatments [34]. It would also be relevant to investigate the factors underlying symptom presence and evaluate interventions that are more precisely tailored to the specific manifestations and characteristics of each case [35,36,37,38].

The high prevalence of symptoms and the negative perception of mental health among medicated patients underscores the need for careful and personalized evaluation of pharmacological treatments. Poorly supported medication combinations and polypharmacy appear to be associated with poorer mental health outcomes in fibromyalgia patients. It is important to recognize that the results of this study are subject to the natural bias of subjective reports from questionnaires and the non-random selection of participants, which limits the findings to those who use social networks. Therefore, it is recommended to compare these findings with results obtained from objective sources.

To implement the prescription and monitoring of medications in fibromyalgia treatment, clinicians should consider developing tailored treatment plans based on individual patient profiles, including symptom severity, mental health status, and comorbidities. Regular assessments using tools like the Fibromyalgia Impact Questionnaire (FIQ) can help monitor treatment efficacy and guide clinical decisions. It is essential to integrate mental health evaluations as part of routine care, as this can directly impact medication adherence and overall treatment success [5]. Clinicians are encouraged to involve patients in shared decision-making, discuss the risks and benefits of pharmacological options, and consider non-pharmacological therapies as complementary strategies [13]. Establishing clear communication channels for patients to report adverse effects promptly can further enhance treatment outcomes and ensure the safe management of fibromyalgia.

Ultimately, these results caution against the prescription of ineffective or counterproductive drug interactions and obligate medical personnel to consider pharmacological management strategies that include close and thorough monitoring, along with the evaluation of complementary treatments that can improve patients’ quality of life without negatively impacting their mental health. Adopting a multidisciplinary and personalized approach to the management of fibromyalgia is essential, as suggested by the literature, to optimize mental health through safe and effective therapeutic strategies. Additionally, patient education on potential side effects and the promotion of non-pharmacological strategies for mental health management could be beneficial [15].

The adverse effects of medications such as benzodiazepines, SSRIs, and anticonvulsants in fibromyalgia patients can be attributed to their mechanisms of action and impact on neurotransmitter systems. Benzodiazepines, for instance, enhance the inhibitory effects of GABA, leading to a generalized suppression of neural activity, which can exacerbate symptoms like cognitive dysfunction and depression commonly seen in fibromyalgia [33]. Similarly, SSRIs, while increasing serotonin levels, can have side effects including emotional blunting and fatigue, which may worsen the overall perception of mental health in patients who already experience mood disturbances.

Moreover, anticonvulsants such as pregabalin modulate calcium channels to reduce neural excitability, aimed at alleviating neuropathic pain. However, this mechanism can also disrupt normal neural signaling, contributing to cognitive and emotional symptoms like memory impairment, anxiety, and depressive mood [8]. The sedative effects of these drugs can reduce motivation and energy levels, further complicating the mental health of fibromyalgia patients.

Future research should focus on understanding the interactions between different medications, dosages, and their long-term impacts on the mental health of fibromyalgia patients, particularly by exploring the minimal pharmacological efficacy needed to support the implementation of multicomponent non-pharmacological treatments that have demonstrated significant benefits not provided by conventional pharmacological treatments. The prescription of promising medications and approaches outside current medical guidelines, such as cannabinoids, vitamin D supplements, and psychedelics like LSD or psilocybin, may offer new therapeutic avenues worthy of in-depth exploration in future studies.

## 5. Conclusions

These findings indicate an association between the use of benzodiazepines and sedative-hypnotics and a negative perception of mental health. While selective serotonin reuptake inhibitors (SSRIs) may reduce the severity of certain symptoms, they do not improve the overall perception of mental health in patients.

Multivariate analysis suggests that drug interactions may contribute to the ineffectiveness of some medications, underscoring the importance of personalizing and optimizing both pharmacological and non-pharmacological treatments. The high prevalence of associated symptoms and negative mental health perceptions emphasizes the need for careful monitoring and integrated therapeutic strategies that combine both approaches to enhance patients’ quality of life without negatively impacting their mental health.

Based on these results, it is evident that non-pharmacological interventions, such as physical exercise, health education, and psychological behavioral techniques, can be effective alternatives for achieving a more comprehensive response in the management of fibromyalgia.

## Figures and Tables

**Table 1 biomedicines-12-02284-t001:** Description of dependent variables.

Variable	Answer Categories	Frequency	Percentage
Self-perception of mental health	Bad	101	18.6
	Fair	293	54.0
	Good	132	24.3
	Very Good	13	2.4
	Excellent	4	0.7
	Total	543	100.0
Prevalence of perceived symptoms	Comprehension or memory problems	524	96.3
	Insomnia	427	78.5
	Depression	401	73.7
	Anxiety	286	52.6
Severity of non-refreshing sleep	Mild	80	14.7
	Moderate	247	45.5
	Severe	216	39.8
	Total	543	100.0
Severity of cognitive symptoms	Mild	225	42.5
	Moderate	247	46.7
	Severe	57	10.8
	Total	529	100.0

Mild = occasional presence, has not been a problem. Moderate = almost always present. Severe = always present, causes major problems.

**Table 2 biomedicines-12-02284-t002:** Drug use and perceived mental health.

Drugs	B	Tip. Error	*p* *	Odds Ratio	95% CI for OR	
					Lower	Upper
GABAergic and anticonvulsants	0.067	0.1742	0.699	1.070	0.760	1.505
SSRI antidepressants	0.276	0.1751	0.115	1.318	0.935	1.857
Analgesics	−0.149	0.1727	0.388	0.862	0.614	1.209
NSAIDs	0.130	0.1873	0.488	1.139	0.789	1.644
Benzodiazepines	−1.018	0.2132	<0.01 *	2.768	1.822	4.204
Muscle relaxants	0.108	0.2105	0.608	1.114	0.737	1.683
Zolpidem/Zopiclone	−0.928	0.2762	<0.01 *	2.531	1.473	4.349
Antidepressants, tricyclics	0.260	0.2652	0.327	1.297	0.771	2.181
Ergotamine (Migranol)	−0.204	0.3047	0.503	0.815	0.449	1.481
Methotrexate	−0.404	0.7945	0.611	0.668	0.141	3.168
Quetiapine	−0.414	0.6521	0.526	0.661	0.184	2.373
Other drug	−0.522	0.3649	0.153	0.593	0.290	1.213

* denotes significant *p*-value < 0.05.

**Table 3 biomedicines-12-02284-t003:** Perception of mental health for pharmacological interactions.

	Benzodiazepines		Benzodiazepines + Zolpidem/Zopiclone		Benzodiazepines + Zolpidem/Zopiclone + SSRI Antidepressant	
	N	%	N	%	N	%
Poor	40	33.3	14	58.3	12	70.6
Fair	65	54.2	6	25.0	3	17.6
Good	12	10.0	2	8.3	2	11.8
Very good	3	2.5	2	8.3	0	0
Total	120	100.0	24	100.0	17	100.0

**Table 4 biomedicines-12-02284-t004:** Effect of drug use on symptomatic presence: logistic regression.

	Cognitive Symptoms			Insomnia			Depression			Anxiety		
Drugs	B(ET)	*p*	OR	B(ET)	*p*	OR	B(ET)	*p*	OR	B(ET)	*p*	OR
GABAergic and anticonvulsants	1.390 (0.577)	<0.02 *	4.016	0.224 (0.236)	0.301	1.276	0.075 (0.219)	0.731	1.078	−0.122 (0.191)	0.524	0.885
SSRI antidepressants	−0.701 (0.566)	0.215	0.496	0.236 (0.235)	0.315	1.266	0.335 (0.218)	0.123	1.398	0.223 (0.190)	0.239	1.250
Analgesics	0.900 (0.585)	0.124	2.459	0.547 (0.235)	<0.02 *	1.728	0.432 (0.216)	<0.05 *	1.540	0.171 (0.186)	0.358	1.187
NSAIDs	−0.121 (0.557)	0.828	0.886	0.683 (0.278)	<0.02 *	1.981	0.454 (0.246)	0.065	1.575	0.151 (0.204)	0.459	1.163
Benzodiazepines	1.407 (1.066)	0.187	4.086	0.824 (0.328)	<0.02 *	2.281	0.628 (0.283)	<0.03 *	1.873	0.551 (0.225)	<0.02 *	1.736
Muscle relaxants	0.797 (0.848)	0.347	2.219	0.423 (0.303)	0.163	1.526	0.254 (0.268)	0.343	1.289	0.042 (0.222)	0.850	1.043
Zolpidem	−0.861 (0.736)	0.242	0.423	0.011 (0.377)	0.977	1.011	0.159 (0.348)	0.647	1.173	0.152 (0.281)	0.589	1.164
Antidepressants, tricyclics	17.500 (4916.783)	0.997	0.000	0.282 (0.394)	0.474	1.325	0.078 (0.344)	0.820	1.082	−0.083 (0.287)	0.774	0.921
Ergotamine (Migranol)	17.583 (5636.134)	0.998	0.000	0.360 (0.470)	0.444	1.434	0.040 (0.391)	0.919	1.041	0.639 (0.346)	0.065	1.895
Methotrexate	18.087 (15,335.32)	0.999	0.000	−0.762 (0.957)	0.426	0.467	0.605 (0.929)	0.515	0.546	0.372 (0.905)	0.681	1.451
Quetiapine	−2.223 (0.935)	<0.02 *	0.108	−0.718 (0.729)	0.325	0.488	0.868 (0.673)	0.197	0.420	0.172 (0.668)	0.796	1.188
Other drug	−0.483 (0.826)	0.559	0.617	−0.411 (0.484)	0.396	1.508	0.413 (0.451)	0.360	1.512	0.706 (0.389)	0.069	2.026
Not used	−0.202 (0.817)	0.805	0.817	−0.348 (0.430)	0.418	1.417	0.037 (0.408)	0.928	1.038	−0.255 (0.399)	0.523	0.775

* denotes significant *p*-value < 0.05.

**Table 5 biomedicines-12-02284-t005:** Perception of severity of cognitive symptoms for pharmacological interactions.

	Benzodiazepines + SSRI Antidepressant + Others		Benzodiazepines + NOT SSRI Antidepressant + Others		SSRI Antidepressant + NOT Benzodiazepines + Others		SSRI Antidepressant + NOT Others	
Severity	N	%	N	%	N	%	N	%
Mild	21	28.4	12	27.3	73	40.1	13	56.5
Moderate	38	51.4	26	59.1	92	50.5	9	39.1
Severe	15	20.3	6	13.6	17	9.3	1	4.3
Total	74	100.0	44	100.0	182	100.0	23	100.0

**Table 6 biomedicines-12-02284-t006:** Perception of severity of non-refreshing sleep for pharmacological interactions.

	Benzodiazepines + NOT Others		Benzodiazepines + Others	
**Severity**	**N**	**%**	**N**	**%**
Mild	0	0.0	11	9.2
Moderate	2	50.0	47	39.5
Severe	2	50.0	61	51.3
Total	4	100.0	119	100.0

## Data Availability

Data used for this manuscript are available upon request.

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
