# Peer review of "Association between Drug Use and Perception of Mental Health in Women Diagnosed with Fibromyalgia: An Observational Study"

_biomedicines, 2024, doi:10.3390/biomedicines12102284_

Round 1

Reviewer 1 Report

Comments and Suggestions for Authors

Abstract

-It would be beneficial to briefly mention the prevalence or impact of fibromyalgia (FM) to contextualize the importance of the study further.

-It might be helpful to explain why a non-probabilistic sampling method was chosen and how it ensures a representative sample, as this is crucial for the external validity of the findings.

-You mention using logistic regression models to evaluate drug use effects on mental health outcomes but do not specify the variables included in the models. Providing a bit more detail about the independent and dependent variables, or at least the main covariates, would enhance the methodological transparency.

-Providing some statistical measures (e.g., odds ratios, confidence intervals) to quantify the impact of drug use on mental health outcomes. This would strengthen the reader's understanding of the magnitude and significance of the findings.

-It would be compelling to briefly suggest what kinds of complementary treatments might be effective, or what specific strategies could be implemented for monitoring.

Introduction

-Expanding slightly to include the demographic most affected by FM (if relevant, such as age or gender predispositions) could further contextualize the significance of the study.

-Central sensitization and other factors like distress and neuroinflammation as possible mechanisms behind FM. Briefly discussing how these mechanisms impact the therapeutic strategies could enhance understanding and link more directly to the treatment options you later discuss.

-Consider discussing the rationale behind the use of these specific medications (e.g., how SNRIs affect neurotransmitters involved in pain and mood regulation) to provide a clearer connection between the pathophysiology and treatment.

-Clarifying why there is a variance in adherence and the consequences of such variances (e.g., increased side effects, decreased effectiveness) could offer deeper insights into the challenges of managing FM.

-Explaining how understanding the impact of medication use on mental health could address some of the treatment adherence or effectiveness issues could strengthen the rationale.

Methods

-A redundancy with the word "prospective" repeated. Removing the duplicate would improve readability.

-It would be helpful to mention if and how the questionnaire was pilot-tested beyond validation to ensure clarity and relevance of the questions for the target population.

-Consider providing a bit more detail about the biopsychosocial dimensions covered by the questionnaire.

-How many rounds of Delphi were conducted and any major changes that arose from this process could provide insight into the robustness and iterative improvement of the questionnaire.

-It would be beneficial to discuss the rationale for choosing a non-probabilistic sample and any potential biases this might introduce, considering the specific challenges of reaching the FM community.

-How participants were verified as having FM (e.g., diagnosis confirmation) when collecting responses would strengthen the reliability of the sample.

-Detailing the assumptions checked before applying logistic regression (such as linearity in the logit for continuous variables) would be beneficial.

-It would be appropriate to also mention any ethical considerations or approvals by an institutional review board or similar body, especially given the nature of the study involving human subjects.

-Adding a sentence about the measures taken to ensure data security during and after data collection would be reassuring for readers and underline the ethical rigor of the study.

Results

-Consider using subheadings for each major finding (e.g., "Impact on Mental Health Perception," "Influence on Symptom Presence," etc.) to make the section easier to navigate.

-Ensure that for each significant finding, the effect size (like odds ratios) and confidence intervals are also reported where applicable.

-If multiple comparisons were made, consider discussing whether any adjustments were made to control for type I error (e.g., Bonferroni correction). This is particularly important in studies with many statistical tests, as it affects the interpretation of significance levels.

Discussion

-Consider expanding this to discuss potential biological or psychological mechanisms in more detail, which could provide a deeper understanding for readers.

-It would be beneficial to provide specific recommendations or guidelines for clinicians on how to implement the suggestions  for careful prescription and monitoring of medications in FM treatment.in everyday clinical practice, such as through the development of tailored treatment plans based on individual patient profiles.

Reviewer 2 Report

Comments and Suggestions for Authors

The manuscript by Lizama-Lefno A et al. entitled “Association between Drug Use and Perception of Mental Health in Women Diagnosed with Fibromyalgia: An Observational Study” is an original and fascinating paper, which is very enjoyable and provides very relevant data, showing that the consumption of drugs not considered in the therapeutic guidelines for fibromyalgia such as benzodiazepines and hypnotics have a significant impact on the patient's perception of their mental health.

The relevance of the manuscript lies in the fact that they show that in fibromyalgia patients, cognitive aspects are affected by the consumption of gabapentin and anticonvulsants, as well as quetiapine; insomnia by analgesics, NSAIDs, and benzodiazepines; depressive symptoms by analgesics, benzodiazepines and anxiety by the consumption of benzodiazepines.

This work is very original, and the results of the association between drugs and symptoms are very relevant from the clinical point of view. It would be very enriching to explain how these drugs achieve the adverse effect observed in one or two paragraphs in the discussion. 

Author Response

Please se the attachment

Round 2

Reviewer 1 Report

Comments and Suggestions for Authors

I have no more comments